# Two Novel *Yersinia pestis* Bacteriophages with a Broad Host Range: Potential as Biocontrol Agents in Plague Natural Foci

**DOI:** 10.3390/v14122740

**Published:** 2022-12-08

**Authors:** Haixiao Jin, Youhong Zhong, Yiting Wang, Chuanyu Zhang, Jin Guo, Xiaona Shen, Cunxiang Li, Ying Huang, Haoming Xiong, Peng Wang, Wei Li

**Affiliations:** 1National Institute for Communicable Disease Control and Prevention, China CDC, Beijing 102206, China; 2State Key Laboratory of Infectious Disease Prevention and Control, Beijing 102206, China; 3Department of Public Health, Medical Institute of Qinghai University, Xining 810016, China; 4Yunnan Institute for Endemic Disease Control and Prevention, Dali 671000, China; 5Yunnan Provincial Key Laboratory for Zoonosis Control and Prevention, Dali 671000, China; 6Qinghai Institute for Endemic Disease Control and Prevention, Xining 811602, China

**Keywords:** *Yersinia pestis*, *Gaprivervirus*, *Myoviridae*, T4-like bacteriophages

## Abstract

Bacteriophages (phages) have been successfully used as disinfectors to kill bacteria in food and the environment and have been used medically for curing human diseases. The objective of this research was to elucidate the morphological and genomic characteristics of two novel *Yersinia pestis* phages, vB_YpeM_ MHS112 (MHS112) and vB_YpeM_GMS130 (GMS130), belonging to the genus *Gaprivervirus*, subfamily *Tevenvirinae*, family *Myoviridae*. Genome sequencing showed that the sizes of MHS112 and GMS130 were 170507 and 168552 bp, respectively. A total of 303 and 292 open reading frames with 2 tRNA and 3 tRNA were predicted in MHS112 and GMS130, respectively. The phylogenetic relationships were analysed among the two novel *Y. pestis* phages, phages in the genus *Gaprivervirus*, and several T4-like phages infecting the *Yersinia* genus. The bacteriophage MHS112 and GMS130 exhibited a wider lytic host spectrum and exhibited comparative temperature and pH stability. Such features signify that these phages do not need to rely on *Y. pestis* as their host bacteria in the ecological environment, while they could be based on more massive *Enterobacteriales* species to propagate and form ecological barriers against *Y. pestis* pathogens colonised in plague foci. Such characteristics indicated that the two phages have potential as biocontrol agents for eliminating the endemics of animal plague in natural plague foci.

## 1. Introduction

Plague is an acute infectious disease caused by *Yersinia pestis* that is mainly found in wild rodents, and parasitic fleas are considered transmitting vectors. To date, public health measures and effective antibiotic treatments have led to a drastic decrease in plague worldwide [1]; however, the disease has still not been eradicated. In fact, more risk for the plague is the endemic threat posed to local residents or travellers in natural plague foci, especially in the background that natural plague foci are distributed widely in the Americas, Asia, and Africa [1]. In addition, multiple drug-resistant strains of *Y. pestis* have been isolated from patients with bubonic plague in Madagascar [2,3]. For streptomycin resistance in *Y. pestis* isolates, plasmid-mediated resistance mechanisms [2,3] and *rpsL* gene mutation mechanisms have also been reported [4,5]. Currently, bacteriophages have been successfully used as disinfectors to kill bacteria in the food, environmental, and agricultural fields and have been used for medical applications in curing human diseases [6].

Many bacteriophages (phages) are capable of lysing *Y. pestis*, and some of them have once been used for the antibacterial treatment of plague [7]. In addition, corresponding phage receptors have been identified in different parts of the LPS core [8,9,10] and/or outer membrane proteins (ail and OmpF) [9]. Several phages have been shown to be highly specifically lytic to *Y. pestis* and are routinely used for the diagnosis of plague. They include lytic phages that resemble members of *Podoviridae* in morphology, such as the Pokrovskaya [11], A1122 [12], Yep-phi [13], etc. A filamentous phage acquired by *Y. pestis*, YpfΦ [14], had the capacity to infect another pathogenic *Yersinia* species (*Yersinia enterocolitica* and *Yersinia pseudotuberculosis*), as well as *Escherichia coli* strains. In addition, *Y. pestis* P2-like temperate phages, such as L-413C (NC_004745) [15], vB_YpM_22, vB_YpM_46, and vB_YpM_50 [16], were also isolated from various environmental backgrounds. A few T4-like phages that can infect *Y. pestis* have been described to date, such as YpsP-PST (KF208315.1) [11], fPS-2 (LR215722), fPS-65 (LR215724), fPS-90 (LR215723) [17], fD1 (HE956711) [18], and JC221 [19].

Two novel *Y. pestis* phages (vB_YpeM_MHS112 and vB_YpeM_GMS130) were isolated in this study. The two phages belong to the genus *Gaprivervirus*, subfamily *Tevenvirinae*, family *Myoviridae*, and the initial characterisations were illustrated using the host range test, one-step growth curve analysis, and stability tests under various stress conditions. The genomes of the two *Gaprivervirus* phage isolates were analysed and compared with phages from the same genus through annotation and alignment. In addition, the corresponding phylogenetic relationships among phages in the genus *Gaprivervirus* and several T4-like phages, including those infecting *Yersinia,* were illustrated in this study.

## 2. Materials and Methods

### 2.1. Isolation and Identification of Phages

Two phages (MHS112 and GMS130) were isolated in two different counties that once belonged to the *Rattus flavipectus* plague focus in Yunnan, China. MHS112 was isolated from the intestinal contents of shrews (*Suncus murinus*) in Menghai County. GMS130 was isolated from stool samples of *R. flavipectus* in Gengma County (Lincang City, Yunnan). Phages MHS112 and GMS130 were isolated with the *Y. pestis* EV strain as the host indicator using similar standard isolation protocols. Briefly, after centrifugation for 10 min at 9000× *g*, the supernatant was filtered through a 0.45 µm filter and incubated with the host strains for 24 h at 37 °C. Luria-Bertani (LB) broth or LB agar plates were used to cultivate *Y. pestis*. SM buffer (5.8 g/L of NaCl, 2.0 g/L of MgSO_4_, 50 mL/L of 1 M Tris, pH 7.5, and 5 mL/L of presterilised 2% gelatine) was used for phage storage and dilutions. Phage plaque assays were performed by the double-layer agar method, as described in [19]. Following appropriate dilution, the suspensions were plated for single plaque formation. Then, after three rounds of single plaque isolation, MHS112 and GMS130 were isolated, and their plaque characteristics were recorded. Phage propagation was performed according to classic procedures [20].

### 2.2. Transmission Electron Microscopy

For electron microscopy, the purified phages were spotted onto 400 mesh carbon-coated grids and negatively stained with 2% phosphotungstic acid (pH 6.5). The grids were observed using a transmission electron microscope (Hitachi HT7700, Tokyo, Japan, 80 kV) with a Gatan 832.10 W CCD camera (Gatan, Pleasanton, CA, USA) operating at Gatan Digital Micrograph software. The dimensions of three viral particles of each phage were measured, and the values were averaged.

### 2.3. Host Range Analysis

Host range analysis was performed using the spot testing method. The host range assays also tested more representative strains of family *Enterobacteriaceae* other than *Y. pestis*, and a total of 114 other genera or species of *Enterobacteriaceae* were included in the host range analysis, including the genera *Yersinia*, *Escherichia*, *Salmonella*, *Shigella, Cronobacter, Klebsiella, Enterobacter, Enterococcus, Cronobacter*, *Serratia, Proteus, Providencia, Morganella, Citrobacter, Pantoea,* etc. (Appendix A).

### 2.4. Growth Characteristics and Thermo, pH Stability Tests

The one-step growth curve of the phage was determined according to the procedures in the previous literature [19] with slight modification. Briefly, 1 mL phage and 1 mL host bacterium (*Y. pestis* EV76 or *E. coli*-CMCC43201) were mixed at an optimal multiplication of infection. The virion samples were taken at different time points (Appendix A), and the titre was measured by the double-layer plate method after serial dilution 10-fold. Three replicates were performed. The measurement of phage temperature and pH stability tests was carried out according to previous literature [21].

### 2.5. Phage Genome Sequencing, Assembly, Annotation, and Comparison

Genomic DNA was extracted from phage propagation picked from a single plaque. Bacterial nucleotides were removed from the phage lysates using DNase 1 and RNase A. Genomic phage DNA was extracted using a Phage Genomic DNA Extraction Kit (ABigen Corp., Beijing, China) and proteinase K according to the manufacturer’s instructions. Phage genomes were sequenced using an Illumina library kit (DNA Library Prep Kit, San Diego, CA, USA) and Illumina NovaSeq PE150 sequencing platform, resulting in >400-fold coverage. The raw sequence data were assembled using SOAPdenovo [22]. The putative genes in phage genomes were identified by the RAST (https://rast.nmpdr.org/, accessed on 14 October 2022) and Galaxy (https://usegalaxy.org/, accessed on 14 October 2022) web services and visualised using Geneious Prime 2022 (https://www.geneious.com, accessed on 21 October 2022). The comparative genetic map was constructed using EasyFig 2.2.5 [23].

### 2.6. Alignments of the Inferred Amino Acid Sequences of gp23

Twenty-two phage genomes in the genus *Gaprivervirus* were acquired from NCBI (Appendix A). In addition, 32 T4-like phage genomes acquired from NCBI were also included in the phylogenetic analysis. The alignments of the inferred amino acid sequences of the structure gene gp23 (major capsid protein) were performed according to previous research [21,24]. BLASTP (https://blast.ncbi.nlm.nih.gov/Blast.cgi, accessed on 18 October 2022) was used to compare the gp23 gene products of MHS112 and GMS130 with other phages in the genus *Gaprivervirus* or T4-like phages [6] (Appendix A). MEGA11 was used to construct the phylogenetic tree [25].

### 2.7. Accession Number

The whole genome sequences of the phages were deposited in GenBank under the accession numbers vB_YpeM_MHS112 (OP750247) and vB_YpeM_GMS130 (OP750248).

## 3. Results

### 3.1. Isolation and Morphology of the MHS112 and GMS130 Phages

The two isolated phages (MHS112 and GMS130) formed clear plaques of 2–3 mm diameter in 24 h in the double-agar layer. According to electron microscopy morphological observations, MHS112 and GMS130 had hexagonal outline heads, indicating their icosahedral symmetry, and had long contractile tails. Such characteristics suggested that they belonged to the family *Myoviridae* under the *Caudovirales* order. The size of the head for MHS112 ranged from 73 to 74 nm in diameter and 93 to 101 nm in length, with tail lengths of 110 nm and 15 nm in width. The size of the head for GMS130 ranged from 59 to 63 nm in diameter and 85 to 89 nm in length, with tail lengths of 85 nm and 15 nm in width (Figure 1). Such elongated heads indicated that the two phages have the characteristics of T4-like phages [18] (coined the genus *Tequatroviruses* according to the taxonomy of prokaryotic viruses updated from the ICTV Bacterial and Archaeal Viruses Subcommittee) [26].

### 3.2. Host Range Determination of MHS112 and GMS130 Phages

MHS112 and GMS130 phage isolates can infect four representative biovars of *Y. pestis* (antiqua, mediaevalis, orientalis, and microtus) in China. MHS112 and GMS130 are also sensitive to *Y. enterocolitica* and *Y. pseudotuberculosis*. In addition, the two phages exhibited relatively wide host ranges in a few species of the family *Enterobacteriaceae,* such as MHS112 and GMS130, which can both lyse *E. coli*, *Shigella sonnei, Salmonella typhimurium,* and *Salmonella blidan.* In the collection of bacterial species in this study, collectively, phage GMS130 showed a broader host range than MHS112 because GMS130 had the ability to infect more non-*Yersinia* strains, including nonpathogenic *E.* coli (ATCC 25,922 and FC 7792), enteroaggregative *E. coli* (EAEC), enterohemorrhagic *E. coli* (EHEC), *Shigella dysenteriae*, *Shigella boydii*, and *Enterobacter cloacae.* Correspondingly, there were some species strains that only MHS112 can infect, such as *Shigella flexneri, E. coli* (ATCC8739, ATCC41446, and MG1655), and *Salmonella cholerasuis* (Appendix A).

### 3.3. Growth Characteristics and Stability Assessment

The one-step growth study revealed that MHS112 and GMS130 had similar latent periods of 20 min, with different burst sizes of approximately 42 and 62 virions per infected bacterium, respectively. When exposed to 28, 37, and 50 °C for 1 h, MHS112 and GMS130 both had high survival rates. The thermostability began to decrease when the temperature was above 60 °C. Assessment of the pH stability test revealed that MHS112 and GMS130 generally survived well after incubation at pH 3–10 in a one-hour period (Appendix A).

### 3.4. Genomic Features of MHS112 and GMS130 Phages

After genome sequencing and data processing, high-quality sequences were obtained. The phage genomes of MHS112 and GMS130 consist of a linear dsDNA sequence, 170,507 (MHS112) and 168,552 (GMS130) in size, and the G+C contents of MHS112 and GMS130 are all 40.5%; this rate is different from that of T4 (34.5%) [27]. Phage MHS112 encodes 303 CDSs, of which 259 lie on the reverse strand and 44 lie on the forward strand. The phage GMS130 encoded 292 CDSs, 247 on the reverse strand and 45 on the forward strand (Appendix A). Two tRNA genes (tRNA-Arg, tRNA-Met) and three tRNA genes (tRNA-Arg, tRNA-Met, tRNA-Asn) were identified in phage MHS112 and GMS130, respectively. MHS112 and GMS130 do not harbour virulence-related genes or antibiotic resistance genes in the two phage genomes, and the absence of lysogenic genes supports the conclusion that they belong to lytic nature phages. The proteins identified for the two phages could be categorised into eleven functional classes [17], i.e., transcription, translation, nucleotide metabolism, virion structure proteins (including head, neck, tail, chaperonins/assembly catalysts), DNA replication/recombination/repair/packaging and processing, lysis, host or phage interactions, host alteration/shutoff, homing endonucleases and homologues, predicted integral membrane or periplasmic proteins, and unknown function hypothetical proteins. The structural and functional arrangement of CDSs is illustrated in Figure 2.

### 3.5. Alignments of MHS112 and GMS130 with Phage SP18 and Phage vB_EcoM_VR20

Genome alignment at the DNA level demonstrated that the phage genomes of MHS112 and GMS130 are 96.77% identical (coverage 87%). In addition, genome comparisons of MHS112 and GMS130 with other representative phages in the genus *Gaprivervirus* revealed that MHS112, at the DNA level, had 95.30% sequence identity to SP18 [28] and 96.61% identity to the low-temperature T4-like coliphage vB_EcoM_VR20 [29], respectively. GMS130 shared 97.35% identity with SP18 and 95.59% identity with vB_EcoM_VR20 (Figure 3A). Comparative genomics revealed that MHS112 and GMS130 phages have very limited nucleotide similarity to previous T4-like (*Tequatrovirus*) phage isolates (JC221) [19] from Yunnan Province (data not shown). In addition, Figure 3B shows the small nonmatched gene cluster (red frame in Figure 3A) containing two genes encoding a long tail fibre distal subunit (gp37) and a chaperone protein (gp38) [17] among MHS112, GMS130, SP18, and vB_EcoM_VR20, and such a mismatch maybe, to some extent, could explain the difference in host lysis ranges.

### 3.6. Phylogenetic Relationship Analysis of Phages in the Genus Gaprivervirus and T4-Like Phages Infecting the Genus Yersinia

The major capture structural protein gp23 genes in the MHS112 and GMS130 genomes all comprise 1560 bp and encode 519 amino acid (aa) residues. Comparing these inferred amino acid sequences of gp23 among phages in the genus *Gaprivervirus* (such as p479 [6], UPEC01, UPEC07 [30], etc.) and several representative T4-like phages in the *Myoviridae* family, including those infecting the genus *Yersinia* T4-like phages (such as YpsP-PST [11], fPS-2, fPS-65, fPS-90 [17], fD1 [18], JC221 [19], etc.). The two novel phage isolates (MHS112 and GMS130) were clustered into the genus *Gaprivervirus*. With the exception of SP18 (a *Shigella* phage), all phages in *Gaprivervirus* use *Escherichia* as a host. In addition, T4-like phages infecting the genus *Yersinia were* mostly grouped into *Tequatrovirus,* with the exception of previously isolated *Y. pestis* phage JC221 [19] from Yunnan Province (Figure 4).

## 4. Discussion

### 4.1. Morphological Properties with T4-Like Phage Characteristics

Because the two phages, MHS112 and GMS130, have elongated heads and contractile tails with base plates (Figure 1), these morphological properties were characterised as T4-like phage properties [18]. Therefore, the two phages in this study were primarily considered T4-like phages (or newly coined as *Tequatrovirus* in subfamily *Tevenvirinae*, family *Myoviridae*). Through genome comparison with other phages in the genus *Gaprivervirus*, MHS112 and GMS130 were found to have above 95% sequence identity with phages in the genus *Gaprivervirus* (Figure 3A). However, there was only 13% collinear nucleotide similarity to the T4 phage (data not shown). According to the classification standard, phage genome sequence similarity greater than 50% is considered the same genus [31], so MHS112 and GMS130 should belong to the genus *Gaprivervirus*, subfamily *Tevenvirinae*, and family *Myoviridae*. Because phages in the genus *Gaprivervirus* all used *E. coli* or *Shigella sonnei* as primer hosts, they were the first phages took *Y. pestis* as one of the hosts and characterised with the genus *Gaprivervirus* properties.

T4-like phages are a diverse group of lytic bacterial *Myoviruses* that share genetic homologies and morphological similarities with the well-studied coliphage T4 [32,33]. In fact, T4-type *Myoviruses* were previously classified into four subgroups based on sequence comparison of three major structural proteins (gp18, gp19 and gp23), i.e., the T-evens, pseudo-T-evens, shizo-T-evens, and exo-T-evens [34]. In addition, with increasing divergence from T4-like phages, the T4-type phage superfamily further comprised at least 15 distinct subgroups [29]. Approximately 90% of the known T4-like phages grow on *E. coli* [24,35,36] or other enterobacteria, but the remaining 10% grow on phylogenetically more distant bacteria (*Aeromonas* [24], *Vibrio* [37,38], cyanobacteria, etc.) [37]. Some T4-like phages infecting the genus *Yersinia* have been identified thus far, such as YpsP-PST [11], phiJA1 [10], fPS-2, fPS-65, and fPS-90 isolated from pig stools in Finland [17]. The phiD1 (fD1) isolated from the sewage of Turku in Finland belongs to the T4-like phage [18]. Previous research found that the phages fPS-2, fPS-65, and fPS-90 also infected *Y. pseudotuberculosis* and *Y. pestis* through LPS and OmpF as receptors [17].

### 4.2. Associated Ecological Function of Phages with Wider Host Ranges

Phages are extremely abundant in the environment and can outnumber their prokaryotic hosts by several orders of magnitude [39]. T4-like phages, as a large, virulent, aggregated phage population, significantly influence microbial ecology and, consequently, affect the entire ecosystem [40]. Phages MHS112 and GMS130 were isolated with *Y. pestis* as the indicator host. However, the two phages exhibited relatively wide host ranges, not only infecting the *Yersinia* genus, such as *Y. pseudotuberculosis* and *Y. enterocolitica*, but also infecting some species in the order *Enterobacteriales,* such as the genera *Escherichia*, *Shigella,* and *Salmonella*.

Many phages are capable of lysing *Y. pestis*. For example, T7-related phages generally have specific lytic characteristics to *Y. pestis*, so they are routinely used for plague diagnostic purposes, such phages as Pokrovskaya [11], A1122 [12], Yep-phi, and YpP-G [7]. However, T7-related virulent phages are not suitable for diminishing the endemicity of animal plague due to their narrow host lysis ranges because once the host bacteria disappear, such phages cannot persist in the environment. For the *Y. pestis* P2-related phages, such as L-413C [15], L-413C was not able to infect 27 *Y. pestis* strains. More importantly, P2-related phages have lysogenic characteristics, and they can coexist with host strains in lysogenic conditions, so P2-related phages are generally not suitable for the purpose of eliminating epizootic endemics in natural plague foci. Similar to T4-like phages (*Tequatrovirus*) in subfamily *Tevenvirinae*, our newly isolated phages MHS112 and GMS130 belong to genus *Gaprivervirus*, subfamily *Tevenvirinae*. The lytic nature endowed the two novel bacterial phages with potential candidates for ecological control in natural plague foci.

### 4.3. The Ecological Barrier Function Endowed Phages with a Wider Host Range in Natural Plague Foci

Our newly isolated *Y. pestis* phages MHS112 and GMS130 exhibited virulent and wider host range characteristics, even though they were primitively isolated with *Y. pestis* as an indicator host. In fact, in the ecological environment, they do not need to rely on *Y. pestis* as their host bacteria, while they could be based on more massive *Enterobacteriales* species to propagate. In addition, the two phages appeared to reproduce more efficiently in *E. coli* strains (CMCC43201) than in *Y. pestis* (Appendix A), similar to results observed in previous research in T4-like phages [18]. Correspondingly, a large number of phages are inhibited in the ecological environment in natural plague foci because they have the capability to lyse *Y. pestis*, so such lysing phages can become an ecological barrier against the *Y. pestis* pathogen colonised in plague foci.

## 5. Conclusions and Perspectives

In this study, two novel *Y. pestis Gaprivervirus* phages (MHS112 and GMS130) were identified, and their genomes were compared with those of closely related phages. The two *Gaprivervirus* phages infect both *Y. pestis* and more *Enterobacteriales* species. Together with comparatively larger burst sizes per infected *Y. pestis* bacterium (approximately 42 and 62 virions per bacterium) and stability characteristics in common factors in the environment, these characteristics indicated that they have potential as microecological biocontrol agents used to diminish the endemicity of plague in natural foci in the future. However, such practice should rely on more scientific research, such as measuring the reduction rate on bacterial load, assessing the inhibition effectiveness in natural plague foci, and developing more practicable phage cocktail schemes, such as more distinct phages, for example, *Tequatrovirus* phages, which were contained in schemes. In addition, by changing various host specificity-associated genes via genetic engineering, phages with a wider host range could be developed, which will greatly benefit from the biocontrol of animal plague in natural plague foci [41].

## Figures and Tables

**Figure 1 viruses-14-02740-f001:**
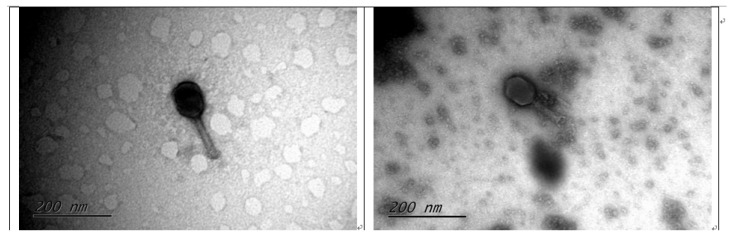
Morphological observations of MHS112 (**left**) and GMS130 (**right**) bacteriophages from transmission electron microscopy.

**Figure 2 viruses-14-02740-f002:**
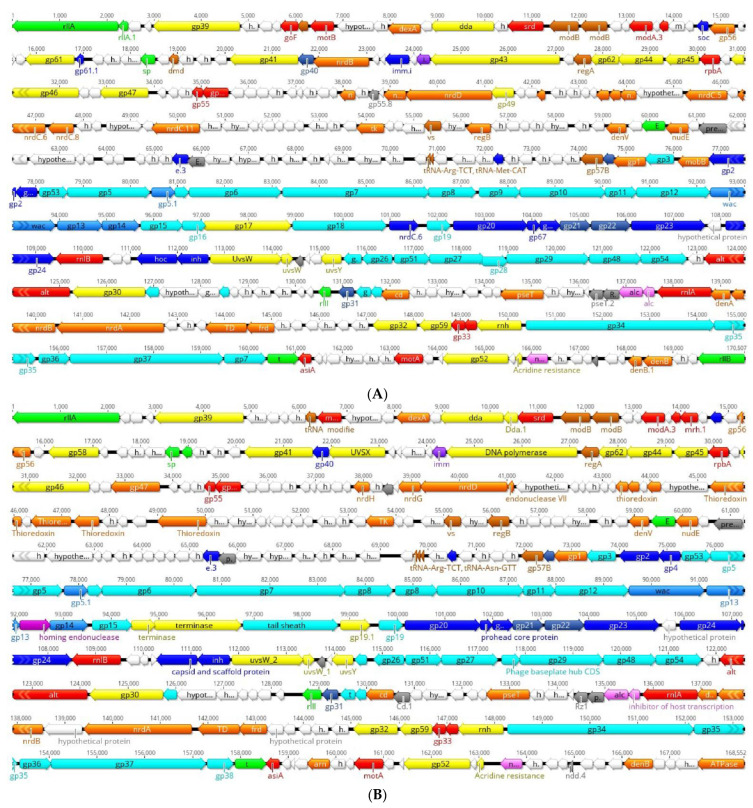
Genomic organisation and structure of *Y. pestis Gaprivervirus* bacteriophages MHS112 (**A**) and GMS130 (**B**). The red and brown arrows indicate transcription- and translation-associated genes, respectively; the green arrows indicate lysis-associated genes; and the yellow arrows indicate genes with corresponding DNA replication, recombination, repair, packaging, and processing functions. The dark blue, medium blue, and light blue arrows indicate the head, neck, and tail genes, respectively, and the stippled blue arrows indicate chaperonins/assembly catalysts genes. The purple and pink arrows indicate host or bacteriophage interactions and host alteration/shut off genes, respectively. The peach arrows indicate homing endonucleases and homologous genes. The grey arrows indicate predicted integral membrane or periplasmic proteins, and the white arrows indicate genes with unknown function.

**Figure 3 viruses-14-02740-f003:**
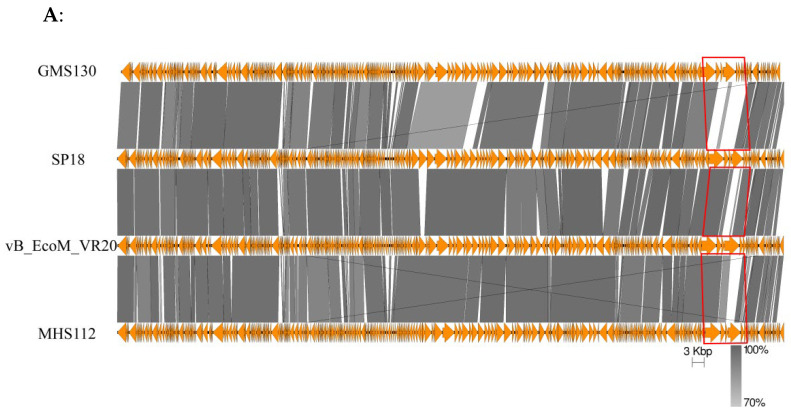
Comparative genomic analysis of bacteriophages MHS112 and GMS130 with bacteriophages SP18 and vB_EcoM_VR20. (**A**) Whole genome alignment of four phage genomes. The red frame indicates a small nonmatched gene cluster. (**B**) Organisation of the small nonmatched gene cluster from the red frame in (**A**).

**Figure 4 viruses-14-02740-f004:**
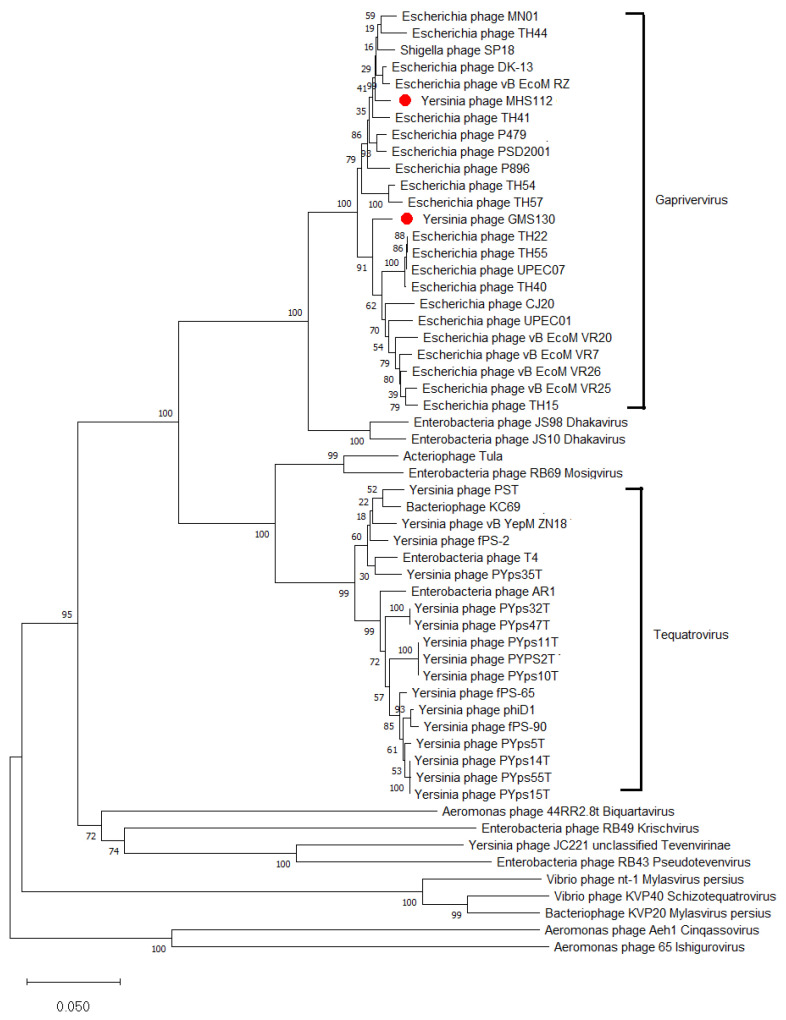
Unrooted phylogenetic tree of the structural gene gp23 in the genus *Gaprivervirus* bacteriophages and representative T4-like phages infecting *Yersinia*.

## Data Availability

Not applicable.

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
