# Peer review of "Two Novel Yersinia pestis Bacteriophages with a Broad Host Range: Potential as Biocontrol Agents in Plague Natural Foci"

_viruses, 2022, doi:10.3390/v14122740_

Round 1

Reviewer 1 Report

Dear Authors,

The manuscript ID: viruses-2058350_v1 entitled Two Novel Yersinia pestis Bacteriophages with a Broad Host Range: Potential as Biocontrol Agents in Plague Natural Foci” is very interesting. In the last years, the antimicrobial resistance has become very widespread and is also one of the most serious global public health threats. Therefore, it is necessary to search for new alternative methods to eliminate pathogenic bacteria, e.g. bacteriophages.

The purpose of this study – isolation, identification and characterization of new phages is a very good idea. The manuscript is properly organized. Introduction contains general data on plaque, Yersinia pestis and Y. pestis phages. Appropriate materials and methods were used to perform the studies. The obtained results are documented, summarized in the form of figures or tables (both in the manuscript and supplementary files) and properly interpreted. Based on the results, discussion and conclusions with perspectives were drawn.

I agree with the Authors that these bacteriophages have potential as biocontrol agents and could be used to reduce plague endemicity in natural outbreaks in the future. However, this requires further research. In my opinion, it is a well written article and may be accepted and published in such a prestigious journal as “Viruses”.

With highest regards,

Author Response

Thank you for your Comments. 

Reviewer 2 Report

The paper by Jin et al reports isolation and preliminary characterization of two Yersinia pestis phage MHS112 and GMS130. It follows a standard phage characterization pathway with results being reported in a cohesive way.

Comments:

1) What is the sequence identity between the two Yersinia phage? Both phage were compared to Shigella phage SP18 (and were shown to be very similar to SP18), however, the similarity between the two phage were not shown. My guess is that they are very similar at the DNA level, given their similarities in size and GC content.

2) If the sequence identity between the two phage is high, what would be the evolutionary pathways between the two phage given that they are isolated in two different regions?

3) What does the sequence similarity like for the gp37 tail fibre distal subunit and gp38 chaperone protein between the two phage? Will the differences in host range between the two phage be explained by the differences in these two proteins?

4) It was commented in the result section that both phage have circularly permuted genomes (line175), how was this determined? What is the evidence for a circularly permuted genome?

5) Figure 2 is very blurry and hard to read. Please revise.

6) It was commented in the conclusion section that both phage have a large burst size (line 319), how was the burst size calculated?

7) Suppl Table 2, “+/- = likely lysis from without”. Please define the term “lysis from without”. What is the mechanism of “lysis from without” for Gram-negative bacteria? How exactly were the experiments performed, and what evidence led to this conclusion?
